# EXTENDING GRAPH TRANSFORMERS WITH QUANTUM COMPUTED AGGREGATION

## ABSTRACT

Recently, efforts have been made in the community to design new Graph Neural Networks (GNN), as limitations of Message Passing Neural Networks became more apparent. This led to the appearance of Graph Transformers using global graph features such as Laplacian Eigenmaps. In our paper, we introduce a GNN architecture where the aggregation weights are computed using the long-range correlations of a quantum system. These correlations are generated by translating the graph topology into the interactions of a set of qubits in a quantum computer. The recent development of quantum processing units enables the computation of a new family of global graph features that would be otherwise out of reach for classical hardware. We give some theoretical insights about the potential benefits of this approach, and benchmark our algorithm on standard datasets. Although not being adapted to all datasets, our model performs similarly to standard GNN architectures, and paves a promising future for quantum enhanced GNNs.

## 1 INTRODUCTION

Graph machine learning is an expanding field of research with applications in chemistry (Gilmer et al., 2017), biology (Zitnik et al., 2018), drug design (Konaklieva, 2014), social networks (Scott, 2011), computer vision (Harchaoui & Bach, 2007), science (Sanchez-Gonzalez et al., 2020). In the past few years, much effort has been put into the design of Graph Neural Networks (GNN) (Hamilton). The goal is to learn a vector representation of the nodes while incorporating information about the graph. The learned information is then processed according to the original problem.

The dominating approach for designing GNNs have been Message Passing Neural Networks (MPNN) (Gilmer et al., 2017). At each layer of MPNNs, a linear layer is applied to the node features, then for each node, the feature vectors of the direct neighbors are aggregated and the result is added to the node feature vector. The way the aggregation is performed differentiates a variety of architectures among which GCN (Kipf & Welling, 2016), SAGE (Hamilton et al., 2018), GAT (Veličković et al., 2018), GIN (Xu et al., 2018).

Despite some successes, it has been shown that MPNNs suffer several flaws. First and foremost, their theoretical expressivity is related to the Weisfeiler-Lehman (WL) test. It means that two graphs who are indistinguishable via the WL test (example in 1a) will lead to the same MPNN output (Morris et al., 2019). This can cause several problems because two different substructures will not be differentiated. This is especially true in chemistry where the graphs represent different molecules (graphs in 1a could represent two molecules). MPNNs also perform best with homophilic data and seem to fail on heterophilic graphs (Zhu et al., 2020). Homophilic graphs mean that two nodes have the same labels if they are close to each other in the graph, which is not necessarily the case. Finally, MPNNs suffer from oversmoothing (Chen et al., 2020) and oversquashing (Topping et al., 2021). Oversmoothing means that the output features of all nodes will converge to the same value as the number of layers increases. Oversquashing occurs when few links on the graph separates two dense clusters of nodes. The information that circulates through these links is then an aggregation of many nodes and is much poorer compared to the information initially present.

Solutions to circumvent those issues are currently investigated by the community. The main idea is not to limit the aggregation to the neighbors, but to include the whole graph, or a larger part of it. Graph Transformers were created in this spirit with success on standard benchmarks (Ying et al., 2021; Rampášek et al., 2022). Similarly to the famous transformer architecture, an aggregation

rule is provided to every pair of nodes in the graph with incorporation of global structural features. Examples of global structural features are Laplacian Eigenmaps (Kreuzer et al., 2021), eigenvectors of the Laplacian matrix.

The goal of this work is to explore new types of global structural features emerging from quantum physics that can be added to a GNN. The rapid development of quantum computers during the previous years provides the opportunity to compute features that would be otherwise intractable. These features contain complex topological characteristics of the graph, and including them could improve the quality of the model, the training or inference time, or the energy consumption.

The paper is organized as follow. Section 2 provides elements about quantum mechanics for the unfamiliar reader and details how to construct a quantum state from a graph. Section 3 provides theoretical insights on why quantum states can provide relevant information that is hard to compute with a classical computer. Section 4 details our main proposal, a GNN architecture with quantum correlations. Section 5 provides a summary on how our work fits in the current literature. Section 6 describes the numerical experiments.

## 2 THE GRAPH AS A QUANTUM SYSTEM

### 2.1 QUANTUM INFORMATION PROCESSING

We provide in this subsection the basics of quantum information processing and quantum dynamics for the unfamiliar reader. More details can be found in (Nielsen & Chuang, 2002; Henriet et al., 2020).

A *quantum state* of a system is a unitary complex vector whose module square of individual entries represent the probability of the system to be in each of these individual entries states. The systems that are often considered are sets of individual two levels systems called *qubits* whose states are denoted $|0\rangle$ and $|1\rangle$. The state of an individual qubit is represented by the complex vector $\begin{bmatrix} \alpha \\ \beta \end{bmatrix}$ with $|\alpha|^2 + |\beta|^2 = 1$, also noted $\alpha |0\rangle + \beta |1\rangle$. A system of $N$ qubits is a vector from $\mathbb{C}^{2^N}$ noted $|\psi\rangle = \sum_{i=0}^{2^N-1} a_i |i\rangle$, $i$ being associated to a bitstring of size $N$. $\left\{ |i\rangle \right\}_i$ is referred as the computational basis of the system. The conjugate transpose of $|\psi\rangle$ is noted $\langle\psi|$.

A quantum state can be modified by an *operator* $U$ which is a complex unitary matrix of size $2^N \times 2^N$. Quantum dynamics follow the Schrödinger equation

$$-i\frac{d|\psi\rangle}{dt} = \hat{\mathcal{H}}(t)|\psi\rangle \tag{1}$$

where $\hat{\mathcal{H}}(t)$ is the *hamiltonian* of the dynamic, a complex hermitian matrix of size $2^N \times 2^N$. The solution to the equation for some time $T$ is $|\psi(T)\rangle = U(T)|\psi(0)\rangle$ where $U(T) = \exp\left[-i\int_0^T \hat{\mathcal{H}}(t)dt\right]$ is the time evolution operator.

Classical information can be extracted from a quantum state by measuring the expectation value of an observable. An *observable* $\hat{\mathcal{O}}$ is a complex hermitian matrix of size $2^N \times 2^N$, and its expectation value on the quantum state $|\psi\rangle$ is the scalar $\langle\psi| \hat{\mathcal{O}} |\psi\rangle$.

We introduce some notations that will be used along the paper. The *Pauli matrices* are the following

$$I = \begin{bmatrix} 1 & 0 \\ 0 & 1 \end{bmatrix}, X = \begin{bmatrix} 0 & 1 \\ 1 & 0 \end{bmatrix}, Y = \begin{bmatrix} 0 & -i \\ i & 0 \end{bmatrix}, Z = \begin{bmatrix} 1 & 0 \\ 0 & -1 \end{bmatrix}$$

A *Pauli string* of size $N$ is a hermitian complex matrix of size $2^N \times 2^N$ equal to the tensor product, or Kronecker product of $N$ Pauli matrices. In the rest of this paper, we will note Pauli strings by their non-trivial Pauli operations and the qubit they act on, with indexing going from right to left. For instance, in a system of 5 qubits, $X_2Y_1 = I \otimes I \otimes X \otimes Y \otimes I$.

## 2.2 Graph Hamiltonians

Given a graph $\mathcal{G}(\mathcal{V}, \mathcal{E})$, we associate a quantum state of $|\mathcal{V}|$ qubits. A hamiltonian of $|\mathcal{E}|$ interactions acting on $|\mathcal{V}|$ qubits with the same topology can be constructed with the form

$$\hat{\mathcal{H}}_\mathcal{G} = \sum_{(i,j)\in\mathcal{E}} H_{ij} \tag{2}$$

where $H_{ij}$ is a hermitian matrix depending on the vertices $i$ and $j$.

We give below several examples of graph hamiltonians respectively the Ising hamiltonian, the XY hamiltonian, and the Heisenberg hamiltonian.

$$\hat{\mathcal{H}}^I = \sum_{(i,j)\in\mathcal{E}} Z_i Z_j \tag{3}$$

$$\hat{\mathcal{H}}^{XY} = \sum_{(i,j)\in\mathcal{E}} X_i X_j + Y_i Y_j \tag{4}$$

$$\hat{\mathcal{H}}^{XXZ} = \sum_{(i,j)\in\mathcal{E}} X_i X_j + Y_i Y_j + J Z_i Z_j \tag{5}$$

with $J > 0$ a constant.

In the examples presented above, the interactions share the topology of the input graph. As such, an evolution with $\hat{\mathcal{H}}_\mathcal{G}$ generates entanglement following the graph connectivity.

## 2.3 Parameterized graph quantum state

With the graph hamiltonians $\hat{\mathcal{H}}_\mathcal{G}$, one can create parameterized quantum states of $|\mathcal{V}|$ qubits containing information about the considered graph. Let $\hat{\mathcal{H}}_M = \sum_{i=1}^{|\mathcal{V}|} X_i$ be a 'mixing' hamiltonian. We then consider quantum states evolving through the Schrödinger equation with hamiltonian

$$\hat{\mathcal{H}}(t) = \Omega(t)\hat{\mathcal{H}}_M + I(t)\hat{\mathcal{H}}_\mathcal{G} \tag{6}$$

where $\Omega(t)$ and $I(t)$ are bounded real functions arbitrarily chosen.

We will call these states *quantum graph states* or *graph states*, not to be confused with the usual definition of graph states in quantum computing (Clark et al., 2005). We consider specifically alternate layered quantum states which can be written

$$|\psi\rangle = \prod_{k=1}^{p} \left( \mathrm{e}^{-i\hat{\mathcal{H}}_M\theta_k} \mathrm{e}^{-i\hat{\mathcal{H}}_\mathcal{G} t_k} \right) \mathrm{e}^{-i\hat{\mathcal{H}}_M\theta_0} |\psi_0\rangle \tag{7}$$

where $\boldsymbol{\theta} = (\theta_0, t_0, \theta_1, t_1, \ldots \theta_p)$ is a real vector of parameters. The choice of these states is motivated by their wide use for quantum algorithms for combinatorial optimization like Quantum Approximate Optimization Algorithm (Farhi et al., 2014), but our procedure can be generalized to other parameterized states depending on a graph Hamiltonian.

## 3 Properties of quantum graph states

Graph quantum states may be able to access topological structures in the graph that would otherwise be very difficult to detect with classical methods. We detail here several examples of graph features that are difficult to compute or learn with classical methods and easy to do with a quantum computer.

### 3.1 Antiferromagnetic states on WL indistiguishable graphs

In figure 1, we provide two graphs and their ground state with respect to the Ising Hamiltonian defined in Eq. (3) ie the state $|\psi\rangle$ minimizing $\langle\psi| \hat{\mathcal{H}}_I |\psi\rangle$. This state will be called the *antiferromagnetic state*, ie the state where a maximum number of neighboring nodes are in opposite states. The two graphs are non distinguishable by the WL test and yet have very different antiferromagnetic states.

## 3.2 XY GRAPH STATES AND CONNECTION TO RANDOM WALKS

We describe here the properties of the $XY$ hamiltonian $\hat{\mathcal{H}}^{XY}$ described in equation 4 that already have been highlighted in (Henry et al., 2021). This hamiltonian has the property to preserve the subspaces with a total occupation number $n$, or the Hamming weight of the bitsring associated to a computational basis state. Let us note $H_n = \text{span}\{ |i\rangle \, |i$ with a Hamming weight of $n\}$. $\hat{\mathcal{H}}^{XY}$ is then block-diagonal, each block acting on one of the $H_n$. The restriction of $\hat{\mathcal{H}}^{XY}$ to $H_1$ is the adjacency matrix of the graph, and performing an evolution starting in $H_1$ is equivalent to make a random walk of a single particle on the graph.

Similarly, an evolution on higher order $H_n$ is equivalent to a $n$ particle random walk with hard-core interactions (i.e. there can be at most one walker per site). Let us define the graph $\mathcal{G}_n$ of size $\binom{N}{n}$, each vertex representing the computational basis states of $H_n$, and $|i\rangle$, $|j\rangle$ being connected if $\langle j| \hat{\mathcal{H}}^{XY} |i\rangle = 1$. An evolution under $\hat{\mathcal{H}}^{XY}$ is equivalent to a random walk on $\mathcal{G}_n$.

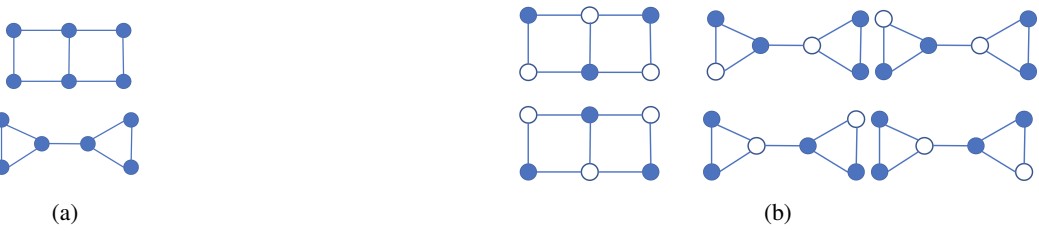

(a)                                                                                      (b)

Figure 1: A pair of graph non distinguishable by the WL test (1a) and associated states minimizing their Ising energy (1b). One of the graph has only ground states with three excited qubits and the other has ground states with two excited qubits (there are also ground states with 3 excited qubits that are not shown here).

## 3.3 NODE CLASSIFICATION ON LATTICE MODELS

In this subsection, we will look at the particular case of lattice graphs and the behavior of classical GNNs on them. A lattice graph is a fully connected subgraph of a regular lattice, containing a full number of cells. In this configuration, several nodes will have identical outputs after going through a MPNN with uniform input features without sampling of neighbors or pooling. Thus there are labelings of nodes that will be impossible to learn with a MPNN for a node classification task.

The phenomenon is illustrated in figure 2, where we show for several lattices (square, triangular, honeycomb and kagome) the outputs of a Graph Convolution Network and an antiferromagnetic node labeling which will not be learnable by a MPNN. This is to be related to previous work showing the limitations of MPNN for heterophilic data, meaning that nodes with the same labels tend not to be neighbors. The proposed labeling can be easily learned with a quantum computer by preparing a quantum graph state with antiferromagnetic correlations. Preparing this state has notably been done with neutral atoms platforms (Scholl et al., 2021).

We also investigate the power of structural embeddings at the core of Graph Transformers like Laplacian Eigenmaps (LE). On each lattice graph, we train a GCN with laplacian eigenmaps as input features and 2 layers for different numbers of input features. The training is done with Adam optimizer for 600 epochs with a .01 learning rate. The results are shown in figure 2c. For all lattices, too few parameters or too few input features results in the failure of the network to learn the labels. 20 input features and 20 hidden neurons are enough to reach a 100% training accuracy. The triangular lattice is the most difficult to learn on. Although the learning is possible with LE features, it may need a substantial amount of parameters to be performed.

## 4 GRAPH TRANSFORMER WITH QUANTUM CORRELATIONS

This section presents our main proposal GTQC (Graph Transformer with Quantum Correlations), an architecture of Graph Neural Network based on Graph Transformers and incorporating global graph features computed with quantum dynamics. A global view of the algorithm is represented on

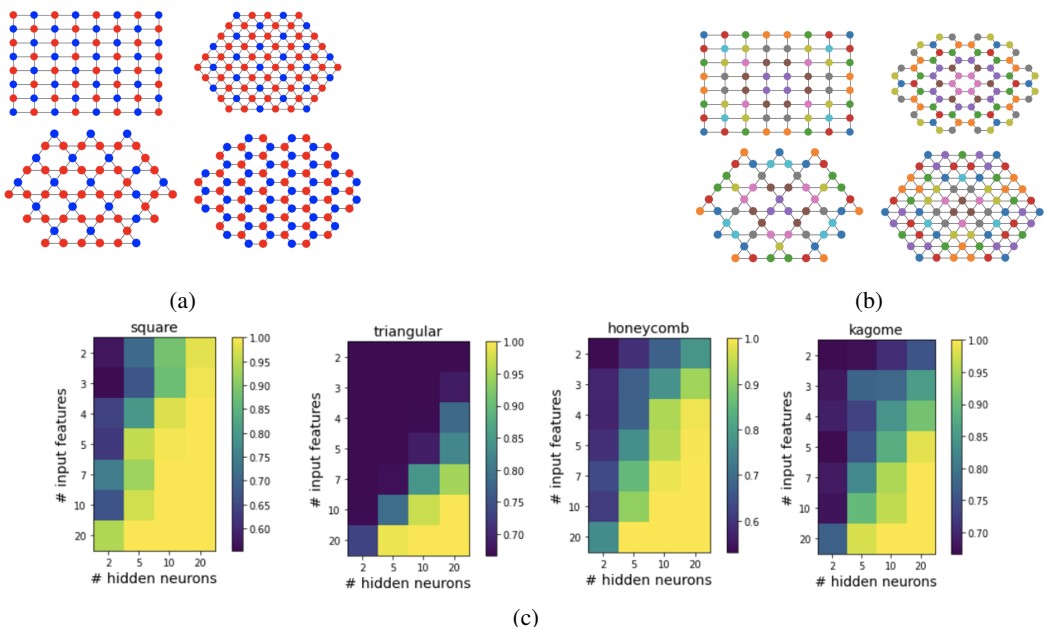

Figure 2: A lattice graph with antiferromagnetic node labeling (2a, not strictly antiferromagnetic for the triangular and kagome lattice but we will name it this way for simplicity) and the output of a Graph Convolution Network (2b). Each color represents a single value of the output of the GCN with uniform node features, 6 layers, 50 hidden neurons per layer. Two nodes with the same color have the same output. Such approach will fail to correctly classify the nodes on the graph with the antiferromagnetic labeling. In 2c is shown the training accuracy of a GCN with laplacian eigenmaps as input features (average over 5 runs).

figure 3. Representation learning on graphs using neural network has become the state of the art of graph machine learning (Wu et al., 2020). Scaling deep learning models has brought lots of benefits as shown by the success of large language models (Brown et al., 2020; Alayrac et al., 2022). The goal was to bring the best of both worlds, meaning large overparameterized deep learning models, and structural graph features intractable with a classical computer. The architecture we propose only uses nodes features, but similar techniques could be implemented for edges features.

## 4.1 TRANSITION MATRIX FROM QUANTUM CORRELATIONS

Here we develop a method to compute a parameterized transition matrix or quantum *attention* matrix from the correlations of a quantum dynamic. It is done with a quantum computer, or Quantum Processing Unit (QPU). This matrix will later be used in the update mechanism of our architecture. Once the quantum attention matrix is computed, the rest of the architecture is purely classical, and all existing classical variations could be implemented. Finally, the quantum attention matrix is by construction equivariant to a permutation of the nodes.

We consider a parameterized graph state as defined in equation 7, parameterized by the trainable parameter $\theta = (\theta_0, t_0, \theta_1, t_1, \ldots \theta_p)$, and noted $|\psi(\theta)\rangle$. $\theta$ will be called the *quantum parameters* in teh rest of the paper. We then compute for every pair of nodes $(i, j)$ the vector of 2-bodies observables $C_{ij} = [\langle Z_i Z_j \rangle, \langle X_i X_j \rangle, \langle Y_i Y_j \rangle, \langle X_i Z_j \rangle, \langle X_i Y_j \rangle, \langle Y_i Z_j \rangle, \langle X_j Z_i \rangle, \langle X_j Y_i \rangle, \langle Y_j Z_i \rangle]^T$ where $\langle O \rangle = \langle \psi(\theta) | O | \psi(\theta) \rangle$.

The quantum attention matrix is computed by taking a linear combination of the previous correlation vector and optionally a softmax over the lines.

$$A(\theta)_{ij} = \gamma^T C_{ij} \tag{8}$$

$$A(\theta)_{ij} = softmax(\gamma^T C_{ij}) \tag{9}$$

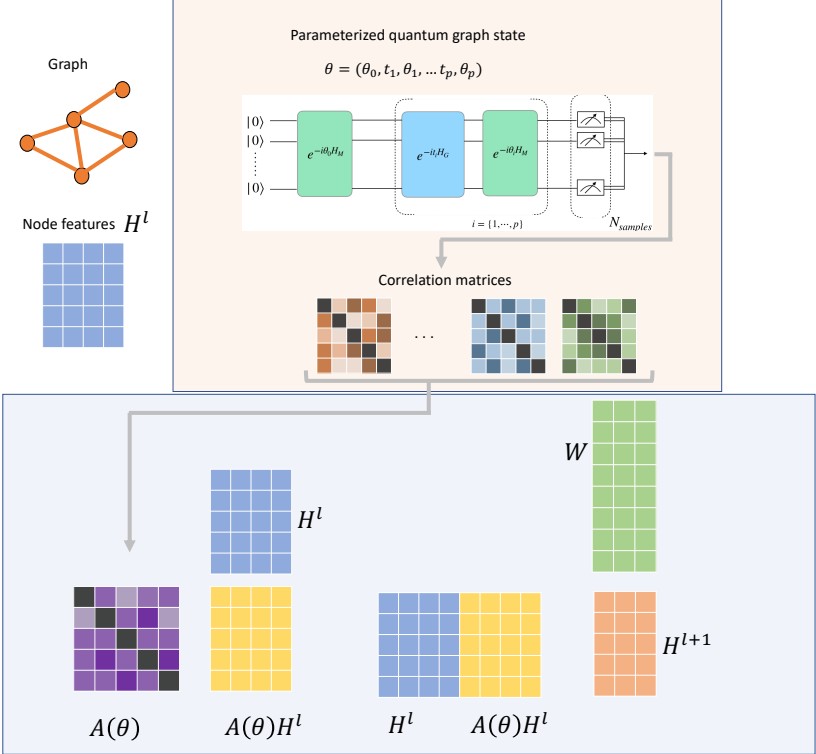

Figure 3: Overview of our model. The graph is translated into a hamitonian that will drive a quantum system. After a parameterized evolution, the correlations are measured and aggregated into a single attention matrix.

where $\gamma$ is a trainable vector of size 9. Multiple correlators are measured to enrich the number of features that can be extracted out of the quantum state. Limiting ourselves to e.g $\langle Z_i Z_j \rangle$ might be inefficient because $\langle \psi_f | Z_i Z_j | \psi_f \rangle$ could be written $X X^\dagger$ where $X$ is a matrix with row $i$ equal to $(Z_i | \psi_f \rangle)^\dagger$. The resulted weight matrix is therefore a symmetric positive semi-definite matrix, and can be reduced by a Choleski decomposition $A = LL^T$ where $L$ is a real matrix of shape $N \times N$. The same model can then be constructed by learning the matrix L even though it is unclear if that would be efficient.

## 4.2 UPDATE MECHANISM

The quantum weight matrix previously computed is used as a transition matrix, or attention matrix, in the update mechanism of our model. Given $H^l$ the node features matrix at the layer $l$, the node features matrix at the next layer is computed with the following formula :

$$H^{l+1} = \sigma((A(\theta)H^l || H^l)W) \tag{10}$$

where $\sigma$ is a non-linearity, $H$ is of size $(N \times d)$ where each row represents a node feature of dimension $d$, $W$ is a learnable weight matrix of size $(2d \times d_h)$, $A(\theta)$ is the attention matrix with parameters $\theta$ computed in 4 and $||$ is the concatenation on the columns.

## 4.3 MULTI HEAD LAYER

With the same approach as Transformers or Graph Attention Networks, one can use several attention heads per layer to improve the expressivity of the architectures. The update formula is given by:

$$H^{l+1} = \Big\|\Big\|_i^{N_{heads}} HEAD_i(H^l) \tag{11}$$

where each head is computed with the formula 4.2. The total dimension of the feature vector is $N_{heads}d_h$. Each head has a different quantum circuit attached, and can be computed in parallel if one possesses several QPUs.

## 4.4 EXECUTION ON A REAL QUANTUM DEVICE

We provide here some precisions on how our model would be implemented on real quantum devices. More details is provided in the supplementary material A.

**Decoupling between quantum and classical parts**
The parameters of the quantum states and the classical weight parameters are independent in our algorithm. One can then asynchronously measure all the quantum states of the model and run the classical part. This may be particularly important for NISQ implementation since the access of QPUs are quite restricted in time. Furthermore, the gradients of the classical parameters depend only on the correlation matrices, so they can be easily computed with backpropagation without any supplementary circuit run.

**Training the parameters of the quantum state**
Computing the gradients of parameterized quantum circuits is a challenge source of numerous research in the quantum computing community (Kyriienko & Elfving, 2021; Wierichs et al., 2022; Banchi & Crooks, 2021). Fortunately some strategies exist and can be deployed, although not for all types of hamiltonians

**Random parameters**
Optimizing over the quantum parameters can be costly and ineffective with current quantum hardware. Even with emulation, back-propagating the loss through a system of more than 20 qubits is very difficult. The idea we propose in the spirit of (Rahimi & Recht, 2008) is to evaluate the attention matrices on many random quantum parameters, and only training the classical weights.

## 5 RELATED WORKS

The use of global graph features has already been used, leading to improvements, in particular positional encoding of the nodes like Laplacian Eigenmaps (LE). It has been shown that a GNN with LE is a universal approximator of the functions over the set of graphs (Kreuzer et al., 2021). However, some functions may require an exponential number of parameters to be approximated.

In recent years, Quantum Machine Learning has seen a fast development with both theoretical and experimental advances Huang et al. (2022); Cerezo et al. (2022). Using quantum computing for Machine Learning on graphs has already been proposed in several works, as reviewed in (Tang et al., 2022). The authors of (Verdon et al., 2019) realized learning tasks by using a parameterized quantum circuit depending on a Hamiltonian whose interactions share the topology of the input graph. Comparable ideas were used to build graph kernels from the output of a quantum procedure, for photonic (Schuld et al., 2020) as well as neutral atom quantum processors (Henry et al., 2021). The architectures proposed in these papers were entirely quantum, and only relied on classical computing for the optimization of variational parameters. By contrast, in our proposal the quantum dynamics only plays a role in the aggregation phase of a larger entirely classical architecture. Such an hybrid model presents the advantage of gaining access to hard-to-access graph topological features through the quantum dynamics, while benefiting from the power of well-known existing classical architectures.

Importantly, the architecture proposed here could be readily implemented on neutral-atom quantum processors (Henriet et al., 2020). In those platforms, the flexibility in the register's geometry enables the embedding of graph-structured data into the quantum dynamics (Henry et al., 2021), with recent experimental implementation. Experimental progress have illustrated the large scalability of those platforms (Schymik et al., 2022), with the realization of arrays comprising more than 300 qubits.

## 6 EXPERIMENTS

### 6.1 TRAINING ON GRAPH COVERS DATASET

We test the performances of our model on a dataset of non isomorphic graphs constructed to be indistinguishable by the WL test. We believe that this task will be especially difficult for classical GNNs and could constitute an interesting benchmark. The way to construct such graphs has been investigated in (Bamberger, 2022) and more details can be found in the supplementary material B. We compare the training loss with one of the most recent implementation of graph transformers (GraphGPS) by (Rampášek et al., 2022). The results are in figure 5. We can see that our model is able to reach much lower values of the loss than GraphGPS, even if both model acheive 100% accuracy after 20 epochs.

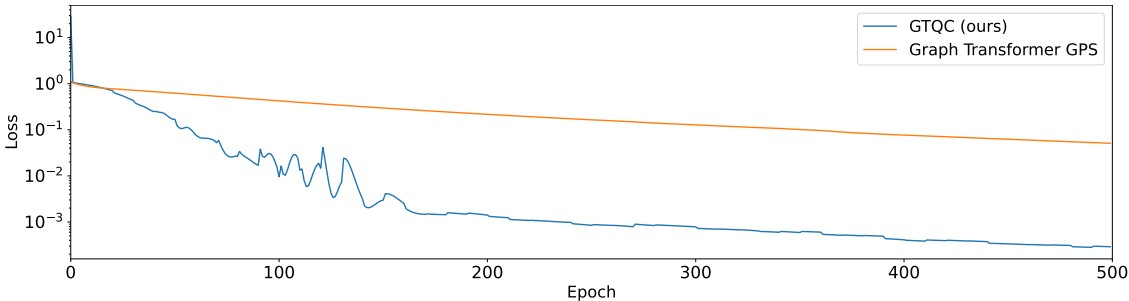

Figure 4: Training loss for our model and a recent graph transformer implementation (GraphGPS) (Rampášek et al., 2022). Both models achieve 100% accuracy after 20 epochs.

### 6.2 BENCHMARK ON GRAPH CLASSIFICATION AND REGRESSION TASKS

We benchmarked our model GTQC and its randomized version of it with different GNN architectures. We selected different datasets from various topics and with diverse tasks to show the general capabilities of our approach. We limited the size of the graphs to 20 nodes in order to be able to simulate the quantum dynamic, and performing a backpropagation. We then chose datasets with the majority of graphs falling below this size limit. The details of each dataset and the protocol can be found in appendix B.

All metrics are such that the lower the better, for classification tasks we display the ratio of misclassified objects.For QM9 the loss is aggregated over all the targets. Figure 5 shows results as box plots accompanied by the underlying points. Though the variability of the results is a bit higher, GTQC reaches similar results than usual classical well-known approaches on QM7 and DBLP_v1 (even beating GCN on this dataset) and seems relevant on QM9 as well. GTQC random usually outperforms GTQC, illustrating the complexity of the optimisation of the quantum system. GTQC random provides very promising results on DBLP_v1 and outperforms all other approaches on QM9. Lettermed seems to represent a difficult task for our quantum methods as they both perform very poorly and way worse than classical methods. QM7 also seems to be challenging for GTQC random. Table 1 shows the results grouped by breadth, compared to the results for GTQC with the same breadth and averaged other various dataset splits. GTQC random has only been trained for breadth of 128 neurons and yields impressive results by clearly outperforming other methods on QM9 and being on par with the best method on DBLP_v1. The raw losses are displayed in the appendix.

## 7 CONCLUSION

We investigated in this paper the opportunity provided by the recent advances in quantum computing hardware to construct new families of graph neural networks. We proposed GTQC, a new architecture of graph transformers that incorporates graph features coming from quantum physics. Specifically, we measure the correlations of a quantum system whose hamiltonian has the same topology the graph of interest. We then use these correlations as an attention matrix for a graph transformer. Although not being adapted to all datasets, our model performs similarly to standard

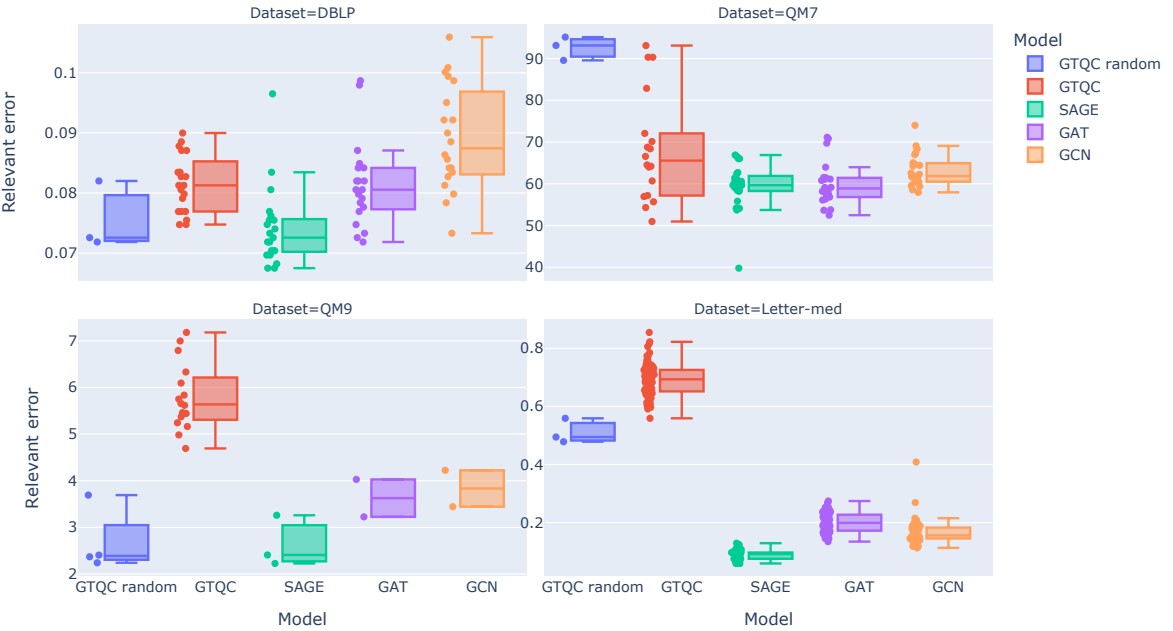

Figure 5: Summary graph of the results of the different models on the studied datasets. Each point is an instance of the model with specific hyperparameters and specific seed for dataset splits.

Table 1: Relative best test error compared to GTQC, grouped by breadth of the hidden layer. We colored in blue the classical models for which at least one of GTQC or GTQC random is better. The standard deviations are over different splits of the datasets.

| breadth | dataset model | DBLP | Letter-med | QM7 | QM9 |
|---|---|---|---|---|---|
| 128 | GAT | -1.74 ± 1.79% | -69.12 ± 3.24% | -6.79 ± 9.27% | -49.70 ± 0.00% |
| | GCN | 6.10 ± 9.51% | -70.55 ± 7.77% | -2.12 ± 6.82% | -47.26 ± 0.00% |
| | SAGE | -9.76 ± 6.12% | -86.59 ± 2.60% | -9.66 ± 5.04% | -59.29 ± 0.00% |
| | GTQC random | -9.41 ± 6.80% | -30.03 ± 0.00% | 34.32 ± 4.09% | -70.79 ± 0.00% |
| | GTQC | 0.00 ± 5.35% | 0.00 ± 8.13% | 0.00 ± 12.32% | 0.00 ± 57.22% |
| 512 | GAT | 4.14 ± 12.31% | -70.58 ± 4.61% | -13.80 ± 7.93% | -48.20 ± 0.00% |
| | GCN | 10.09 ± 9.94% | -75.93 ± 3.01% | -9.69 ± 3.82% | -44.72 ± 0.00% |
| | SAGE | -10.45 ± 4.44% | -86.34 ± 2.73% | -12.31 ± 4.37% | -61.33 ± 0.00% |
| | GTQC | 0.00 ± 5.77% | 0.00 ± 6.25% | 0.00 ± 26.36% | 0.00 ± 41.08% |
| 1024 | GAT | 1.10 ± 12.66% | -69.44 ± 5.53% | -17.38 ± 4.08% | * |
| | GCN | 9.65 ± 16.50% | -77.96 ± 2.63% | -7.24 ± 4.27% | * |
| | SAGE | -13.51 ± 3.20% | -87.22 ± 1.76% | -9.65 ± 4.53% | -57.73 ± 0.00% |
| | GTQC | 0.00 ± 8.21% | 0.00 ± 8.52% | 0.00 ± 24.73% | 0.00 ± 2.80% |
| 2048 | GAT | -1.28 ± 5.01% | -74.00 ± 5.01% | -9.55 ± 5.08% | * |
| | GCN | 14.60 ± 9.63% | -80.13 ± 2.65% | -7.37 ± 3.51% | * |
| | SAGE | -3.13 ± 13.44% | -88.07 ± 2.13% | -16.96 ± 9.80% | * |
| | GTQC | 0.00 ± 3.80% | 0.00 ± 7.65% | 0.00 ± 3.78% | * |

GNN architectures on different benchmarks. While the exact capabilities of our approach are not yet well understood, quantum enhanced GNNs are a promising family of models that could be fully exploited with near term quantum hardware.

## 8 REPRODUCIBILITY STATEMENT

The code to reproduce all the experiments is available in the supplementary material, with the description of all seeds and hyperparameters.

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

## A  EXECUTION ON A REAL QUANTUM DEVICE

**Training the parameters of the quantum state**
Computing the gradients of parameterized quantum circuits is a challenge source of numerous research in the quantum computing community (Kyriienko & Elfving, 2021; Wierichs et al., 2022; Banchi & Crooks, 2021). Finite-difference methods fail because of the sampling noise of quantum measurements and the hardware noise. Some algorithms named parameter-shift rules were then created to circumvent this issue (Mitarai et al., 2018). In some cases, the derivative of a parameterized quantum state can be expressed as an exact difference between two other quantum states with the same architecture and other values of the parameters.

We detail here how we would compute the gradient in a simple case of our architecture. Let $\hat{\mathcal{H}}$ be a hamiltonian, $\hat{\mathcal{O}}$ an observable, $|\psi_0\rangle$ an initial state. We introduce

$$|\psi(\theta)\rangle = U(\theta)|\psi_0\rangle = \exp(-i\theta\hat{\mathcal{H}})|\psi_0\rangle \tag{12}$$

$$f(\theta) = \langle\psi(\theta)|\,\hat{\mathcal{O}}\,|\psi(\theta)\rangle \tag{13}$$

It is known (Wierichs et al., 2022) that $f$ can be expressed as a trigonometric polynomial

$$f(\theta) = \sum_{\omega\in\Omega} a_\omega \cos(\omega\theta) + b_\omega \cos(\omega\theta) \tag{14}$$

where $\Omega$ is a finite set of frequencies depending on the eigenvalues of $\hat{\mathcal{H}}$. In the case of $\hat{\mathcal{H}} = \hat{\mathcal{H}}_M$, the frequencies are the integers between 0 and $N$. One can then evaluate $f$ on $2N+1$ points and solve the linear equations to determine $\{a_\omega\}$, $\{b_\omega\}$ and the derivative of $f$. This is the same for the $\hat{\mathcal{H}}^I$ hamiltonian which associated frequencies are the integers between 0 and $|\mathcal{E}|$.

For the other hamiltonians listed, there is no known analytical formula to compute the gradients like previously. The only method would be to use finite difference schemes which are sensitive to noise.

**Alternate training**
At the time of this work, quantum resources are very expensive, so we want to limit ourselves in the number of access to the QPU. One way to do so is not to update every parameter at each epoch. Typically the gradients of the quantum parameters are expensive to compute so the update would be less frequent. Due to the decoupling between the quantum and classical parameters, one is able to compute the gradients of the weights matrix with only the quantum attention matrices stored in memory.

**Random parameters**
Optimizing over the quantum parameters can be costly and ineffective with current quantum hardware. Even with emulation, back-propagating the loss through a system of more than 20 qubits is very difficult. We encounter memory errors for more than 21 qubits on our A100 GPUs, even though our implementation is certainly not optimal. Therefore we propose an alternative scheme to our model, to help with both actual hardware implementations and classical emulation.

The main idea in the spirit of (Rahimi & Recht, 2008) is to evaluate the attention matrices on many random quantum parameters, and only training the classical weights. From a model $f(x;W,\theta) = \Big\|\Big|_i^{N_{heads}} \sigma((A(\theta_i)H^l||H^l)W_i)$ with one layer, we would normally find the parameters that minimize a loss between inputs $x$ and labels $y$

$$W^*, \theta^* = \arg\max_{W,\theta} \sum_{i=1}^{M} l(f(x_i;W,\theta), y_i) \tag{15}$$

Instead, we create a model with more heads and fixed random values. $\theta'$ expressed as $f(x;W,\theta') = \Big\|\Big|_i^{N'_{heads}} \sigma((A(\theta'_i)H^l||H^l)W_i)$ and we minimize only on $\theta$

$$W^* = \arg\max_{W} \sum_{i=1}^{M} l(f(x_i;W,\theta'), y_i) \tag{16}$$

## B SUPPLEMENTARY INFORMATION ABOUT EXPERIMENTS

### B.1 DETAILS ABOUT DATASETS USED

#### B.1.1 QM7 AND QM9 MOLECULES AND GRAPH REGRESSION

**Context**
QM7 dataset is a subset of the GDB-13 database (Blum & Reymond, 2009), a database of nearly 1 billion stable and synthetically accessible organic molecules, containing up to seven heavy atoms (C, N, O, S). Similarly QM9 is a subset of the GDB-17 database consisting of molecules with up to nine heavy atoms. Learning methods using QM7 and QM9 are predicting the molecules electronic properties given stable conformational coordinates.

**QM7 figures**
QM7 consists of 7165 molecule graphs. Each node is an atom with its 3D coordinates and atomic number Z. The only edge feature is the entry of the Coulomb matrix. Each graph is thus fully connected and has one regression target corresponding to its atomization energy.

**QM9 figures**
QM9 consists of 130831 molecule graphs of between 1 and 29 nodes with an average of 18 nodes (see Figure 6). Each node is an atom with its 3D coordinates and its atomic number Z. Edges are purely distance based and have no feature. Each graph is thus fully connected and has 12 regression targets corresponding to diverse chemical electronic properties. In our implementation, all the targets are recentered and rescaled by their standard deviation.

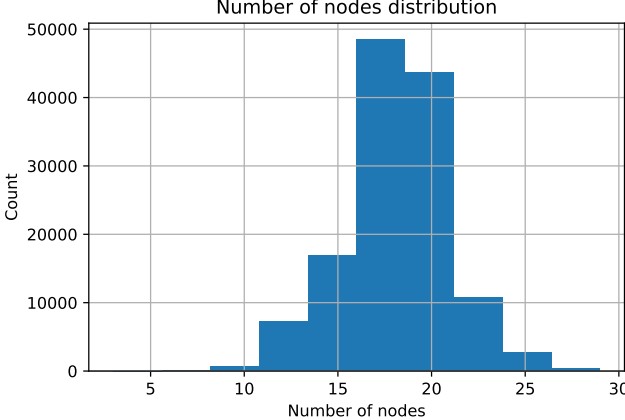

Figure 6: Distribution of the number of nodes in a graph for QM9 dataset.

**Benchmarks**
On the QM7 dataset, Quantum Machine benchmark reached MAE of 3.5 and 9.9 (Montavon et al., 2013).

The best models of the MoleculeNet benchmark (Wu et al., 2017) reached a test MAE of 2.86 ± 0.25 on QM7 and 2.4 ± 1.1 on QM9.

#### B.1.2 DBLP_v1 AND NODE CLASSIFICATION

**Context**
DBLP_v1 is a graph stream built out of the DBLP dataset (Pan et al., 2013) consisting of bibliography data in computer science. To build a graph stream, a list of conferences from DBDM (database and data mining) and CVPR (computer vision and pattern recognition) fields are selected (as shown in Table 2). The papers published in these conferences are then used (in chronological order) to form a binary-class graph stream where the classification task is to predict whether a paper belongs to DBDM or CVPR field by using the references and the title of each paper.

| Conf. category | Conferences | #Papers | #Graphs |
|---|---|---|---|
| DBDM | SIGMOD, VLDB, ICDE, EDBT, PODS, DASFAA, SSDBM, CIKM, DEXA, KDD, ICDM, SDM, PKDD, PAKDD | 20601 | 9530 |
| CVPR | ICCV, CVPR, ECCV, ICPR, ICIP, ACM Multimedia, ICME | 18366 | 9926 |

Table 2: DBLP_v1 details.

Papers without references are filtered out. Then, the top 1000 most frequent words (excluding stop words) in titles are used as keywords to construct the graph (see Figure 7).

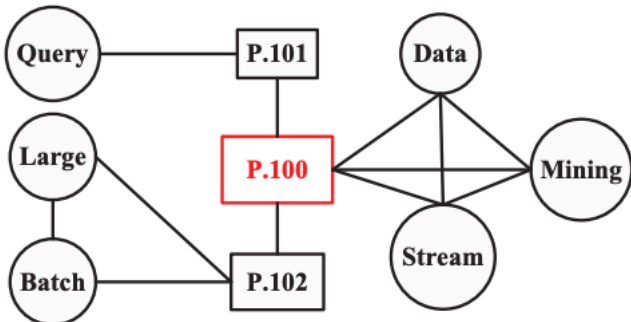

Figure 7: Graph representation for a paper (P.100) in the DBLP_v1 dataset. The rectangles are paper ID nodes and circles are keyword nodes from titles. The paper P.100 cites (connects) paper P.101 and P.102, and P.100 has keywords Data, Stream, and Mining in its title. Paper P.101 has keyword Query in its title, and P.102's title include keywords Large and Batch. For each paper, the keywords in the title are linked with each other

**Figures**
DBLP_v1 consists of 19456 graphs evenly split between the two groups of conferences (the two classes) from 2 to 39 nodes with an average of 10 nodes.

These graphs are actually local parts of a bigger graph. To perform node classification (on nodes representing a paper), the local neighborhood of each graph is extracted and a graph classification task is run.

There are 3 types of edges:

- 0: paper - paper
- 1: keyword - paper
- 2: keyword - keyword

Node features are only a unique ID to be identified between multiple graphs (cf P.100 in Figure 7 which will also appear in P.101 graph). There are 41325 unique IDs so keywords don't have a single keyword identifier among all graphs.

**Benchmarks**
DBLP_v1 (Pan et al., 2013) benchmarks show accuracy ranging between 0.55 and 0.80 in a chunked graph stream classification setup.

**Data augmentation**
The map from IDs to topics and paper IDs is also provided and has been used to perform data augmentation to provide node features. USing Stanford GloVe word embedding pre-trained on

Wikipedia 2014 and Gigaword 5 in a 50-dimension space, each topic node could be enriched with its embedding. A boolean flag was also added to identify a node as a topic or not (a paper).

### B.1.3 COMPUTER VISION: LETTERS AND GRAPH CLASSIFICATION

**Context**
Letters datasets (Riesen & Bunke, 2008) are 3 datasets of distorted letter drawings with low, medium or high distorsion levels (see Figure **??**). Only the 15 capital letters of the Roman alphabet that consist of straight lines (A, E, F, H, I, K, L, M, N, T, V, W, X, Y, Z) are represented. Distorted letter drawings are converted into graphs by representing lines by undirected edges and ending points of lines by nodes.

**Figures**
Node attributes are their 2D positions and edges have no attribute. The graphs are uniformly distributed over the 15 letters. We focused on the medium distorsion dataset consisting of 2250 graphs.

**Benchmarks**
Benchmark results from k-NN are given by (Riesen & Bunke, 2008): 99.6% (low), 94.0% (medium), and 90.0% (high). The best classical algorithm we trained on this dataset was GraphSAGE with results of 100% (low), 94.5% (medium), and 80% (high)).

### B.1.4 GRAPHCOVERS AND THE WEISFEILER-LEHMAN ISOMORPHISM TEST

**Context**
As explained previously and illustrated in Section 3 and Figure 1, Graph Neural Networks expressivity is related to the Weisfeiler-Lehman (WL) test. Recent work has been made to generate datasets of graphs undistinguishable by the WL test (Bamberger, 2022).

**Figures**
Using this work, we generated a small dataset of 6 non-isomorphic graphs of 21 nodes that can't be distinguished by GNNs. These 6 graphs are then split in 3 arbitrary classes. The task at hand consists in being able to correctly distinguish. the classes on the train data.

**Benchmarks**
Even on the train data, as they can't distinguish between the graphs, usual GNNs are not able to learn as shown in (Bamberger, 2022).

### B.2 PROTOCOL

We implemented the models with a classical emulator of quantum circuits implemented in *pytorch* (Paszke et al., 2019) and with *dgl* (Wang et al., 2019), and ran them on A100 GPUs. All experiments were done using one GPU, except QM9 which required 4. We used a Adam optimizer with a learning rate .001, no weight decay for 500 epochs. The quantum parameters were only updated every 10 epochs because of computation time. As an order of magnitude, one epoch of QM7 takes 6 min with 1 GPU, and one epoch of QM9 takes 1h with 4 GPUs. Most of the time is allocated to compute the quantum dynamic, the size of classical parameters has little effect on the compute time except for the 2048 hidden layers.

We compare our model to three architectures of message passing models : GCN (Kipf & Welling, 2016), SAGE (Hamilton et al., 2018), GAT (Veličković et al., 2018). In order to have a fair comparison between the models, we employ similar hyperparameters for all of them. Each model has 2 layers and 1 head for multi-head ones like ours and GAT. We also report the results by comparing models that have the same number of hidden neurons per layer and therefore approximately the same number of parameters. Each dataset is randomly split in train, validation, test with respective ratios of .8, .1, .1 and the models are run on 5 seeds, except QM9 for which we have only one seed.

### B.3 RAW LOSSES FOR EACH MODEL

Table 3: Raw losses for each model

| model | dataset breadth | DBLP | Letter-med | QM7 | QM9 |
|---|---|---|---|---|---|
| GAT | 128 | 0.08 ± 0.00 | 0.22 ± 0.02 | 64.27 ± 6.40 | 4.03 ± 0.00 |
|  | 512 | 0.08 ± 0.01 | 0.20 ± 0.03 | 60.42 ± 5.56 | 3.23 ± 0.00 |
|  | 1024 | 0.08 ± 0.01 | 0.21 ± 0.04 | 56.06 ± 2.77 | 0.00 ± 0.00 |
|  | 2048 | 0.08 ± 0.00 | 0.19 ± 0.04 | 59.27 ± 3.33 | 0.00 ± 0.00 |
| GCN | 128 | 0.09 ± 0.01 | 0.21 ± 0.05 | 67.50 ± 4.70 | 4.22 ± 0.00 |
|  | 512 | 0.09 ± 0.01 | 0.16 ± 0.02 | 63.30 ± 2.67 | 3.44 ± 0.00 |
|  | 1024 | 0.09 ± 0.01 | 0.15 ± 0.02 | 62.94 ± 2.90 | 0.00 ± 0.00 |
|  | 2048 | 0.09 ± 0.01 | 0.14 ± 0.02 | 60.70 ± 2.30 | 0.00 ± 0.00 |
| GTQC | 128 | 0.08 ± 0.00 | 0.71 ± 0.06 | 68.95 ± 8.50 | 8.01 ± 4.58 |
|  | 512 | 0.08 ± 0.00 | 0.66 ± 0.04 | 70.09 ± 18.48 | 6.23 ± 2.56 |
|  | 1024 | 0.08 ± 0.01 | 0.67 ± 0.06 | 67.85 ± 16.78 | 5.27 ± 0.15 |
|  | 2048 | 0.08 ± 0.00 | 0.73 ± 0.06 | 65.53 ± 2.48 | 5.30 ± 0.45 |
| GTQC random | 32 | 0.00 ± 0.00 | 0.56 ± 0.00 | 0.00 ± 0.00 | 0.00 ± 0.00 |
|  | 64 | 0.00 ± 0.00 | 0.48 ± 0.00 | 0.00 ± 0.00 | 0.00 ± 0.00 |
|  | 128 | 0.08 ± 0.01 | 0.49 ± 0.00 | 92.62 ± 2.82 | 2.34 ± 0.09 |
| SAGE | 128 | 0.08 ± 0.01 | 0.09 ± 0.02 | 62.30 ± 3.47 | 3.26 ± 0.00 |
|  | 512 | 0.07 ± 0.00 | 0.09 ± 0.02 | 61.47 ± 3.06 | 2.41 ± 0.00 |
|  | 1024 | 0.07 ± 0.00 | 0.09 ± 0.01 | 61.30 ± 3.07 | 2.23 ± 0.00 |
|  | 2048 | 0.08 ± 0.01 | 0.09 ± 0.02 | 54.41 ± 6.42 | 0.00 ± 0.00 |

