# OpenReview forum: "Extending graph transformers with quantum computed aggregation"
_ICLR.cc/2023/Conference — Submitted to ICLR 2023_

### Official Review · Reviewer_FV73 · 2022-10-21

**Confidence:** 2
**Correctness:** 3
**Technical Novelty And Significance:** 2
**Empirical Novelty And Significance:** 1
**Recommendation:** 5

**Clarity, Quality, Novelty And Reproducibility:**

Clarity: This paper provides good preliminaries for readers unfamiliar with quantum computing, which was helpful. However, this paper is not clear in what contribution it brings to the machine learning community and how the contribution is empirically validated.

Novelty: This paper newly develops a self-attention-like architecture for graphs, which is novel. However, the novelty is not very clear in the text since the related works are only briefly mentioned.

**Strength And Weaknesses:**

Strength 1: Designing neural network architectures for quantum processing is an active & significant area of research.

Strength 2: To my knowledge, this paper is the first to propose quantum computer-version of graph transformers.

Weakness 1: While the paper mainly focuses on introducing new global graph features, the benefit of introducing additional feature, e.g., better expressive power or empirical performance, is not clear.

Weakness 2: The motivating example is not very convincing since I think the example (samples from antiferromagnetic Ising model) can be obtained using existing graph neural network. Samples (or labels) from antiferromagnetic Ising model can be framed as samples from a probabilistic graphical model (PGM). Obtaining such a sample has been well-studied in the context of structured prediction, e.g., graph Markov neural network.

Weakness 3: The paper does not empirically compare with the existing quantum graph neural network, hence the contribution of this paper as a “new quantum neural network for graphs” is not clear.

Weakness 4: This paper only computes the attention matrix using quantum processing units and requires CPU or GPU for other operations. It is not clear how the whole architecture can be efficiently implemented in practice.

**Summary Of The Paper:**

This paper proposes a new graph transformer architecture based on quantum computation. The main idea is to replace the typical attention matrix by a “quantum computable” attention matrix. The main motivation for such a proposal is easily getting access to graph features (such as maximum probability assignment from an antiferromagnetic Ising model) using quantum computation. The proposed algorithm is simulated under small scale experiments (graphs with up to 20 nodes) and shows comparable performance when compared to popular non-quantum coputation-based graph neural networks.

**Summary Of The Review:**

This paper proposes a new quantum computing-based architecture for graphs. Although I am a non-expert in quantum computing, I do not think this paper brings clear contribution to the machine learning community in current form.

---

> ### Author Response · Authors · 2022-11-18
> **Reply to the reviewer**
>
> We thank the reviewer for the feedback, and we adress some clarifications on the weaknesses.
>
> Weakness 1:
> We agree that new theoretical and experimental work should be carried out to show the full benefits of our method. This is currently limited by computational capabilities, and more work should also be done to make classical emulations more efficient before quantum hardware is fully available.
>
> Weakness 2:
> We only looked how difficult it would be for classical GNNs to learn an antiferromagnetic labeling (fig 2). We show that MPNNs with uniform features could not learn such embeddings due to the symmetry of the system, and we look at the number of parameters required for MPNNs with positional encoding features to learn such a labelling. We show in the latter case that although learnability is possible, a substantial amount of parameters is needed. In comparison, it is easy to prepare such states with a quantum computer (https://www.nature.com/articles/s41586-021-03585-1). As part of our future work, we will focus on designing this figure better so that its meaning is clearer.
>
> Weakness 3:
> Our paper proposes a fundamentally different structure than the existing quantum graph neural networks. Our model is hybrid quantum-classical, with the output being classical node features whereas in other quantum graph neural networks all the computation is done on the quantum computer, the classical computer being only used for the optimization part. One could integrate easily the outputs of our model on current machine learning pipelines.
>
> Weakness 4:
> It is true that an important engineering challenge will have to be made in order for the model to be effectively ran, but important progress has been recently made in the building of quantum hardware.

---

### Official Review · Reviewer_fcGg · 2022-10-23

**Confidence:** 4
**Correctness:** 2
**Technical Novelty And Significance:** 3
**Empirical Novelty And Significance:** 1
**Recommendation:** 3

**Clarity, Quality, Novelty And Reproducibility:**

The content and the structure of this article is clear. In this paper, quantum computation is combined with traditional GNN method to form quantum enhanced GNN, which not only retains the framework of traditional GNN, but also introduces graph quantum computation. The method is novel. But the experimental results are not convincing. In general, the quality of this paper is average.

**Details Of Ethics Concerns:**

For the part of Experiment 6.1, can you compare the performance of traditional GNN or transformer with the model proposed in this paper in terms of structure detection under general rather than specific design data sets. Or design new experiments or experimental indicators to prove the advantages mentioned in Section 3.1.  In addition, can you further propose and explain other advantages of the model proposed in this paper from a theoretical or experimental perspective.

**Strength And Weaknesses:**

**Strengths：**
- This paper extends graph transformers with quantum computed aggregation and introduce a GNN architecture where the aggregation weights are computed using the long-range correlations of a quantum system. The method this paper uses is novelty.
- The goal of this paper is to explore new types of global structural features emerging from quantum physics that can be added to a GNN. And the author explains the reasons for adopting graph quantum state from three aspects. This makes the motivation clear.
- The whole paper is clear in structure and content.

**Weaknesses：**

The author mentioned that this method can easily detect some complex structures compared with traditional methods, and gave some special cases in Chapter 3, which is the motivation of the author to introduce the graph quantum states, but these special cases can not show that the graph structure detection ability of quantum graph GNN is greater than that of traditional methods. From the aspect of structure detection, this paper lacks the relevant experimental comparison with traditional methods in general situations, rather than the carefully constructed data in Section 6.1. If you want to show that the graph quantum state has better structure detection capability than traditional methods on some graphs, both two models in Section 6.1 have reached 100% accuracy, and the loss index based on this may not be convincing. At the same time, according to the results of comparative experiments on various tasks in Section 6.2, it is difficult to explain the advantages of this method compared with traditional methods, which is not enough to support the author's motivation. In addition, this paper does not prove other advantages of GTQC.

**Summary Of The Paper:**

In this paper, a new quantum enhanced GNN is proposed from the perspective of quantum mechanics. The paper regards the graph as a quantum system, transforms the graph topology into the interaction of a group of qubits, and uses the long-range correlation of the quantum system to calculate the relationship between different vertex to form the aggregation weight of GNN. The author illustrates the advantages of graph quantum states compared with traditional methods to obtain graph structures from three aspects. By rewriting the update mechanism of traditional GNN, the proposed model can distinguish some graph structures that are difficult to distinguish by traditional GNN. The experimental results show that this new model performs similarly to standard GNN architectures on different benchmarks.

**Summary Of The Review:**

The experiments in this paper fail to prove the possible advantages of quantum graph GNN mentioned in Section 3. At the same time, the author does not give other capabilities of the model, which leads to the ambiguity of the meaning of the designed model. From this point of view, this paper should be rejected.

---

> ### Author Response · Authors · 2022-11-18
> **Reply to the reviewer**
>
> We thank the reviewer for the feedback. He clearly understood the motivations and the contributions of the paper. We agree that new theoretical and experimental work should be carried out to show the full benefits of our method. This is currently limited by computational capabilities, and more work should also be done to make classical emulations more efficient before quantum hardware is fully available.

---

> > ### Comment · Reviewer_fcGg · 2022-12-06
> > **Response to author feedback**
> >
> > Thanks for the author's reply. Considering most concerns still remain, I would maintain my original ratings

---

### Official Review · Reviewer_w3FW · 2022-10-23

**Confidence:** 4
**Correctness:** 3
**Technical Novelty And Significance:** 2
**Empirical Novelty And Significance:** 1
**Recommendation:** 3

**Clarity, Quality, Novelty And Reproducibility:**

The quantum part is not easy to follow for a basic-knowledge person. I have asked some domain experts for help understanding the technical part. The experimental part is not very solid and needs improvements.

**Strength And Weaknesses:**

Pros:
1.	It’s interesting to combine quantum computing with Graph Neural Networks.
2.	The authors show another way to construct an attention map, which may bring some inspiration.

Cons:

[Whether the requirements.are realizable]

The proposed algorithm requires a large reliable quantum computer. The authors claimed that their algorithm outperforms the 1-WL test, which is illustrated by finding the ground state for some particular physical system. However, finding the ground state of some strongly correlated systems is proven intractable for classical computers (i.e., NP-hard [1]). Thus, it is hard to say that their algorithm also works well on a classic computer in practical applications.

[No theoretical guarantee on classic computer].

It’s great to design an algorithm that only works well on a quantum computer and enjoys better expressiveness. However, in the current stage, using a reliable quantum system is not realistic. Given this, the authors need to justify that their model outperforms GNN state-of-the-art variants (e.g., ESAN[2]). Therefore, the significance of the algorithm is questionable.

[1] F. Barahona, J. Phys. A 15, 3241 (1982)
[2] Beatrice Bevilacqua et al. Equivariant subgraph aggregation networks. In International Conference on Learning Representations, 2022.

[Experimental results]

The author compared their model with several out-of-date baseline methods but showed superior performance. The method needs improvement, and the authors should include strong baselines for comparisons, such as TorchMD-Net and ESAN.

**Summary Of The Paper:**

This paper proposed a quantum computing-based Graph Transformer architecture. The authors illustrate that their model outperforms the 1-WL test by finding the ground state energy of the Ising Hamiltonian. They also implement their algorithm on the classical computer and conduct experiments on some classical graph tasks. Experimental results show that their model is comparable to other GNN architectures.

**Summary Of The Review:**

The way of improving graph transformers is interesing but the current quality of the work, especially the empirical results, are not ready to publish in the venue, and I recommend rejection.

---

> ### Author Response · Authors · 2022-11-18
> **Reply to the Reviewer**
>
> We thank the reviewer for the feedback and the rightful questioning of the applicability of our approach and our experiments.
>
> [Whether the requirements.are realizable]
>
> Our method is designed to run on an actual quantum computer, and the experiments are made with emulators of quantum systems. Although it is true that most of current quantum systems are small and noisy, some neutral atoms analog systems currently operate in the regime of classical non simulatability (see for example https://www.nature.com/articles/s41586-021-03582-4 and https://www.nature.com/articles/s41586-021-03585-1). Our algorithm with random correlations could already be implemented on those systems with little adaptation, the only thing missing would be the ability to measure XY or XZ correlators.
>
> [Experimental results]
>
> We compared our model to simple baselines in order to have close comparisons. The goal was to get as close as possible to well understood models while changing only the aggregation rule part. We felt it was a better way to compare the expressivity of the different models than to be fine-tuning hyper-parameters and lose the generality of the approach. The benchmarks we propose are in all cases limited, because we cannot evaluate numerically our model on graphs of more than 20 nodes. We welcome the reviewer’s suggestion to include more up-to-date baselines and leave it as a future step of our work.

---

### Decision · Program_Chairs · 2023-01-20

**Decision:**

Reject

**Justification For Why Not Higher Score:**

All reviewers agreed that the work is not good enough or marginally below the acceptance threshold. Authors also agreed that new theoretical and experimental work should be carried out to show the full benefits of [their] method. These items were not addressed during the rebuttal / discussion period.

**Justification For Why Not Lower Score:**

The paper has merits as outlined above.

**Metareview: Summary, Strengths And Weaknesses:**

This work investigates GNNs and new types of global structure features from quantum physics that could be added to a GNN.
Strengths are the novelty of the explored topics and clarity of writing. Weaknesses are the insufficient theoretical and experimental work to show the benefits of the method. In the responses the authors agree with some of the main perceived weaknesses and indicate they have several ideas how the work could be further improved.

In view of the reviews and discussion, I conclude that this article makes promising initial contributions but that it could be improved in important ways, and in particular it would have been stronger by providing more theoretical and experimental evidence in support to the possible benefits of the proposed method.


**Summary Of Ac-Reviewer Meeting:**

NA